# Selection of Vaccine Candidate for Foot-and-Mouth Disease Virus Serotype O Using a Blocking Enzyme-Linked Immunosorbent Assay

**DOI:** 10.3390/vaccines9040387

**Published:** 2021-04-15

**Authors:** Yimei Cao, Kun Li, Xiangchuan Xing, Huifang Bao, Nana Huang, Guoqiang Zhu, Xingwen Bai, Pu Sun, Yuanfang Fu, Pinghua Li, Jing Zhang, Xueqing Ma, Dong Li, Zaixin Liu, Zengjun Lu

**Affiliations:** 1State Key Laboratory of Veterinary Etiological Biology, OIE/National Foot-and-Mouth Disease Reference Laboratory of China, Lanzhou Veterinary Research Institute, Chinese Academy of Agricultural Sciences, Lanzhou 730046, China; caoyimei@caas.cn (Y.C.); likun02@caas.cn (K.L.); xxc15232058169@163.com (X.X.); baohuifang@caas.cn (H.B.); h1540414396@163.com (N.H.); zhuguoqiang1998@126.com (G.Z.); baixingwen@caas.cn (X.B.); sunpu@caas.cn (P.S.); fuyuanfang@caas.cn (Y.F.); lipinghua@caas.cn (P.L.); zhangjing@caas.cn (J.Z.); maxueqing@caas.cn (X.M.); lidong@caas.cn (D.L.); liuzaixin@caas.cn (Z.L.); 2Tianjin Weite Biological Medicine Co., Ltd., Tianjin 300301, China

**Keywords:** foot-and-mouth disease, serotype O, NA-ELISA, VNT, vaccine selecting

## Abstract

Foot-and-mouth disease (FMD) is a highly contagious disease and one of the most economically important diseases of livestock. Vaccination is an important measure to control FMD and selection of appropriate vaccine strains is crucial. The objective of this study was to select a vaccine candidate and to evaluate the potential of a blocking ELISA for detecting neutralizing antibodies (NA-ELISA) in vaccine strain selection. Binary ethylenimine inactivated vaccines, prepared from four representative circulating strains (FMDV O/Mya/98, SCGH/CHA/2016, O/Tibet/99, and O/XJ/CHA/2017) belonging to four lineages within three different topotypes of FMD virus (FMDV) serotype O in China, were used to vaccinate cattle (12–13 animals for each strain), sheep (12–13 animals for each strain), and pigs (10 animals for each strain). The results of immunogenicity comparison showed that O/XJ/CHA/2017 exhibited the highest immunogenicity among the four strains in pigs, cattle, and sheep both by NA-ELISA and virus neutralizing test (VNT). Cross-neutralization analysis indicated that O/XJ/CHA/2017 displayed broad antigen spectrum and was antigenically matched with other three representative strains both by NA-ELISA and VNT. In addition, A significant correlation (*p* < 0.0001) was observed between the NA-ELISA titers and the VNT titers for four representative strains. The results showed that O/XJ/CHA/2017 was a promising vaccine strain candidate and NA-ELISA was comparable to VNT in neutralizing antibodies detection and could be used as the reference test system for vaccine strain selection.

## 1. Introduction

Foot-and-mouth disease (FMD) is a highly contagious viral disease affecting cloven-hoofed animals such as cattle, pigs, sheep, goats, and about 70 species of wild animal. It leads to enormous economic losses due to severe reduction in animal productivity, high mortality in newborn animals and trade restriction on affected countries [1,2]. In areas (Asia, and Africa) where FMD is endemic, disease control still relies mainly on vaccination with inactivated vaccines. The causative agent, FMD virus (FMDV), is a single-stranded, positive-sense RNA virus belonging to the *Aphthovirus* genus of the family Picornaviridae. The virus exists as seven distinct serotypes (A, O, C, Asia 1, SAT1, SAT2, and SAT3) with each serotype containing multiple and constantly evolving strains [3,4]. Among the seven serotypes of FMDV, serotype O is the most widely distributed in the world.

Serotype O includes 11 genetic topotypes, namely Middle East–SouthAsia (ME-SA), Southeast Asia (SEA), Indonesia-1 (ISA-1), Indonesia-2 (ISA-2), Cathay, West Africa (WA), East Africa (EA-1, EA-2, EA-3, and EA-4), and Europe-South America (Euro-SA) [5]. The SEA topotype, the Cathay topotype, and the ME-SA topotype containing the PanAsia lineage and the IND2001 lineage, are currently prevalent in China. There is no cross-immunity between serotypes and lack of cross-immunity between some strains within a serotype. Consequently, FMDV-specific antibodies protect only against homologous, but not heterologous FMD outbreaks. Thus, the vaccine selected must be highly specific to the strain involved and matched as closely as possible with the circulating field isolate. In addition, it is preferable that vaccines cover a broad antigenic spectrum to increase the level of cross-protection [6].

O/ME-SA/IND2001 lineage of FMDV was not detected in China before 2015; however, in February 2017, an outbreak of FMDV infection occurred in Kashgar region in Xinjiang, Northwest China. The representative isolate was identified to belong to O/ME-SA/IND2001 lineage d and named as O/XJ/CHA/2017 (GenBank No. MF461724) by World Reference Laboratory for FMD (WRLFMD) at the Lanzhou Veterinary Research Institute (LVRI), Lanzhou, China. The nucleotide identity of VP1 coding region sequences between O/XJ/CHA/2017 and FMDV O/Mya/98 was only 82.6% [7], and FMDV O/Mya/98 is used for vaccine strain in China [8,9]. The low nucleotide identity of O/XJ/CHA/2017 with this vaccine strain implies a potential spread risk of FMDV O/ME-SA/Ind2001 lineage viruses in China. Therefore, it is necessary to evaluate the current isolates to select appropriate vaccine strains.

The gold standard for FMD vaccine matching is in vivo vaccination-challenge experiments, but the live viral challenge tests have limitations from the perspective of animal welfare, biosafety, and economics. In practice, vaccine selection is mainly based on in vitro serological vaccine matching test, i.e., virus neutralization tests (VNT) and liquid-phase blocking ELISA (LPBE). VNT is more relevant to in vivo protection than other measures [10] and has been widely used for many years [11]. However, it is labor intensive, requires live virus and the results are inconsistent due to the use of different cells and different interpretations of cytopathic effects (CPE). Although LPBE has the advantages of being fast, simple, easy to scale up and not requiring live virus, traditional polyclonal antisera based LPBE has low specificity [12,13] and is difficult to standardize. In addition, strain-specific guinea pig and rabbit antibodies are needed to improve accuracy.

A blocking ELISA for detecting neutralizing antibodies (NA-ELISA) against FMDV serotype O based on bovine broadly neutralizing monoclonal antibodies (bnAbs) was developed previously in our lab [14], it has high sensitivity and specificity, and could detect neutralizing antibodies against FMDV serotype O strains of three topotypes with a wide antigenic and molecular diversity [14]. In this study, the inactivated vaccines prepared from four representative circulating strains (FMDV O/Mya/98, SCGH/CHA/2016, O/Tibet/99 and O/XJ/CHA/2017) were used to vaccinate cattle, sheep, and pigs. We compared the immunogenicity of FMDV O/Mya/98, SCGH/CHA/2016, O/Tibet/99 and O/XJ/CHA/2017 vaccines in pigs, cattle, and sheep, and selected one with highest immunogenicity for cross-neutralizing analysis using NA-ELISA and VNT, with the aim of selecting a potential vaccine candidate and evaluating the potential application of NA-ELISA in vaccine strain selection.

## 2. Materials and Methods

### 2.1. Ethics Statement

All animal experiments were performed following the management guidelines of the Gansu Ethical Review Committee (license no. SYXK-89 GAN-2014-003). All animals used in the present study were humanely bled.

### 2.2. Serum Samples

One hundred and thirty-five serum samples with known protective results against FMDV O/Mya/98 (GenBank No. JN998085) were provided by vaccine manufacturer, which were collected from 135 pigs vaccinated with different doses of the FMDV serotypes O (FMDV O/Mya/98) inactivated vaccine at 28 days post-vaccination (dpv) prior to the challenge and were used to understand the relationship between NA-ELISA antibody titers and clinical protection. The antibody titers against FMDV O/Mya/98 were determined using NA-ELISA and the protection percentage of antibody positive (serum with a titer of ≥1.35) and negative (a titer of <1.35) animals were calculated. These pigs were used in vaccine potency assays, so received different vaccine dose.

### 2.3. Cells and Viruses

Baby hamster kidney (BHK)-21 adherent cells were cultured in Dulbecco’s modified Eagle medium (Thermo Fisher Scientific, Waltham, MA, USA). The media were supplemented with 10% fetal bovine serum (pH 7.4), and the cells were grown at 37 °C in a 5% CO_2_ incubator. Low-passage FMDV O/Mya/98 belonging to Mya-98 lineage in SEA topotype, SCGH/CHA/2016 (GenBank No. KX161429) belonging to Cathay topotype, O/Tibet/99 (GenBank No. AJ539138) belonging to PanAsia lineage in ME-SA topotype and O/XJ/CHA/2017 (GenBank No. MF461724) belonging to India 2001 lineage in ME-SA topotype obtained from WRLFMD at the LVRI and were serially passaged two to three times in adherent BHK-21 cells. Stocks of virus were prepared by infecting BHK-21 cell monolayers and were stored as clarified tissue culture harvest material at −70 °C until required. The above virus strains were used for the VNT, NA-ELISA and vaccine preparation.

### 2.4. Vaccine Preparation

The FMDV O/Mya/98, SCGH/CHA/2016, O/Tibet/99 and O/XJ/CHA/2017 were grown in BHK-21 monolayer and harvested after observing complete CPE. The virus supernatant was inactivated with 3 mM binary ethylenimine at 30 °C for 28 h. The supernatant was neutralized by treatment with sodium thiosulfate. The inactivated virus was concentrated with PEG6000 and purified by sucrose density gradient ultracentrifugation to estimate the antigen content in the concentrated suspension by spectrophotometer at 259 nm. The vaccines were formulated with 146S antigen and Montanide ISA 201 (Seppic, Shanghai, China) by homogenization to have 6 μg antigen per dose in 2 mL vaccine.

### 2.5. Animal Vaccination

Forty 2-month-old white pigs, fifty 10-month-old cattle, and fifty 12-month-old sheep, verified as negative for FMDV neutralizing antibody (virus-neutralizing antibody titer < 1:4), were purchased locally and kept in the healthy animal stalls of the LVRI of the Chinese Academy of Agricultural Sciences (CAAS). Pigs, cattle, and sheep were randomly divided into four groups, each group was inoculated with 2 mL of vaccine SCGH/CHA/2016, O/Mya/98, O/Tibet/99 or O/XJ/CHA/2017 at day 0, 21, respectively. Animal grouping and vaccine immunization were detailed in Table 1. Serum samples were prepared and pooled for each group on 21 and 42 days post primary vaccination (dpv).

### 2.6. Virus Neutralizing Test

Sera from vaccinated animals were assayed with four representative strains (O/Mya/98, SCGH/CHA/2016, O/Tibet/99, and O/XJ/CHA/2017) from the three topotypes of FMDV serotype O by using VNT on monolayers of BHK-21 cells as previously described [15]. Briefly, serum samples were 2-fold serially diluted in 96-well cell culture plates in a total volume of 50 μL, and then 100 TCID_50_ of FMDV in 50 μL of culture media was added to each well. After incubation for 1 h at 37 °C, 5 × 10^4^ BHK-21 cells in 100 μL media were added to each well as indicators of residual infectivity. Normal cell wells, and 0.1, 1, 10, and 100 TCID_50_ virus control wells were used in each plate. The plates were incubated at 37 °C under 5% CO_2_ conditions for 72 h before fixing and staining. The endpoint titers were calculated as the reciprocal of the last serum dilution to neutralize 100 TCID_50_ FMDV in 50% of the wells. Titers of ≥1.65 and ≤0.9 were considered positive and negative, respectively.

### 2.7. NA-ELISA

Sera from vaccinated animals were also assayed with above four representative strains by using NA-ELISA as previously described [14]. Briefly, serum samples were tested in 96-well plates in 2-fold dilutions. Eight sample dilutions (from 1:4 to 1:512) were incubated overnight at 4 °C with a pretitrated dose (1 µg/mL) of FMDV serotype O 146S antigen in a saline buffer liquid phase (final dilution, 1:8 to 1:1024). Subsequently, 100 µL of the serum-antigen mixtures was transferred to an ELISA plate that had been coated with 1 µg/mL of bovine bnAbs F128 in a 100-µL volume and preblocked with blocking buffer (5% sucrose and 1% BSA in PBS), and then the plate was incubated at 37 °C for 1 h. After five washes with PBS containing 0.05% Tween (vol/vol) (PBST), 2 µg/mL of biotinylated bovine bnAbs E46 (Bio-E46) in a 100-µL volume was added to each well, and then the plate was incubated at 37 °C for 1 h. After five washes, 100 µL of 1:30,000-diluted HRP-conjugated streptavidin was added, and the plates were incubated at 37 °C for 15 min. After five washes with PBST, 100 µL of the enzyme substrate TMB was added to each well, and then the plate was incubated at 37 °C for 10 min to 15 min. The reaction was terminated with 2 M H_2_SO_4_, and the optical density (OD) was measured using an automatic microplate reader (BioTek, Winooski, VT, USA) at a wavelength of 450 nm. Four wells were used for antigen control (100% reactivity), and two wells were used as reaction blanks without the 146S antigen and without serum. Antibody titers were expressed as the reciprocal (log10) of the serum dilutions giving 50% of the absorbance recorded in the antigen-control wells (OD50). Serum with a titer of ≥1.35 and a titer of <1.35 was considered positive and negative, respectively. Both F128 and E46 were bovine bnAbs against FMDV serotype O produced previously in our lab using a combination of fluorescence-activated cell sorter (FACS) and techniques for the isolation of single B cells [14].

### 2.8. The r1-Value Determination

VNT and NA-ELISA antibody titers obtained from serum samples collected from cattle inoculated with the highest immunogenicity vaccine at 42 dpv were used to calculate r1-values using the formula: r1 = log10 serum titer against heterologous strain/log10 serum titer against homologous strain. r1-values ≥0.3 were interpreted as cross-protective, and r1-values < 0.3 as non-protective.

### 2.9. Statistical Analysis

Antibody titers were presented as the mean ± standard deviation (SD). Unpaired *t*-test with Welch’s correction was used to compare the differences between the groups. Pearson’s coefficient test was used to determine the correlation between the NA-ELISA titers and the VNT titers. Statistical analysis was performed using GraphPad Prism (version 6.0) software (San Diego, CA, USA), and *p* value of <0.05 was considered statistically significant. In figures, *p* value criteria were the following: * *p* < 0.05, ** *p* < 0.01, *** *p* < 0.001, **** *p* < 0.0001.

## 3. Results

### 3.1. Comparison of Immunogenicity in Cattle, Sheep, and Pigs

The immunogenicity of FMDV O/Mya/98, SCGH/CHA/2016, O/Tibet/99, and O/XJ/CHA/2017 vaccines were compared in pigs, cattle, and sheep after the first vaccination by NA-ELISA (Figure 1A). FMDV O/XJ/CHA/2017 produced significantly higher titers of homologous antibodies (*p* < 0.01, *t*-test) compared with FMDV O/Mya/98 at 21 dpv in pigs, the mean antibody titer was the highest among the four vaccines, reaching 2.2 log10 (SD = 0.4), the mean antibody titer (2.4 log10, SD = 0.4) induced by FMDV O/XJ/CHA/2017 was equivalent to those induced by O/Tibet/99 (2.4 log10, SD = 0.2) in cattle, which were higher than those induced by the other two vaccines. In sheep, the antibody titer produced by O/XJ/CHA/2017 vaccine was also the highest, with a mean of 2.0 log10 (SD = 0.3) (Figure 1A). After the second vaccination, the NA-ELISA antibody titers were substantially increased in all the vaccinated groups except Group G (cattle vaccinated with O/Tibet/99) at 42 dpv (Figure 1B). Similarly, the mean antibody titer produced by O/XJ/CHA/2017 strain was the highest among the four strains all in pigs, cattle, and sheep (Figure 1B).

VNT for homologous viruses was performed on sera collected at 42 dpv from all animals vaccinated with FMDV O/Mya/98, SCGH/CHA/2016, O/Tibet/99 and O/XJ/CHA/2017 vaccines. The mean neutralizing antibody titer produced by O/XJ/CHA/2017 vaccine was the highest among the four vaccines all in pigs, cattle, and sheep (Figure 1C). Taken together, both the VNT and NA-ELISA results indicated that O/XJ/CHA/2017 strain showed the highest immunogenicity among the four strains all in pigs, cattle, and sheep. Therefore, O/XJ/CHA/2017 strain was selected as a possible vaccine candidate for further analysis.

### 3.2. Cross-Neutralization Analysis

NA-ELISA for heterologous virus strains was performed on serum samples collected from pigs (Group D), cattle (Group H) and sheep (Group L) vaccinated with O/XJ/CHA/2017 vaccine at 21 dpv and 42dpv. The antibody titer for all three heterologous viruses was significantly different from that of homologous virus (*p* < 0.05, *t*-test) in pig (Figure 2A) and sheep (Figure 2C) at 21 dpv, but the antibody titer for only one heterologous virus SCGH/CHA/2016 was significantly different from that of homologous virus (*p* < 0.05, *t*-test) in cattle (Figure 2B) at 21 dpv. The mean antibody titer for homologous virus was relatively low at the first vaccination, so the antibody titer for heterologous virus was lower. However, there were still two heterologous viruses (FMDV O/Mya/98 and O/Tibet/99) with mean antibody titers greater than 1.35 all in pigs, cattle, and sheep (Figure 2A–C). After the boost vaccination, the antibody titer for heterologous virus O/Mya/98 was not significantly different from that of homologous virus (*p* > 0.05, *t*-test) both in pig (Figure 2A) and cattle (Figure 2B) at 42 dpv, but there were still significant differences between all the heterologous antibody titers and homologous antibody titer in sheep at 42 dpv (*p* < 0.05, *t*-test; Figure 2C). Although the antibody titers for heterologous viruses were significantly different from those of homologous virus in sheep at 42 dpv, the mean antibody titer for heterologous virus SCGH/CHA/2016 with the greatest difference also reached 1.6 log10 (>1.35). The results indicated that the O/XJ/CHA/2017 vaccine exhibited a good antigen spectrum.

VNT for heterologous viruses was performed on serum samples collected from pigs, cattle and sheep vaccinated with O/XJ/CHA/2017 vaccine at 42 dpv. the same viruses described above in NA-ELISA were used. The VN antibody titer for heterologous virus O/Mya/98 was not significantly different from that of homologous virus (*p* > 0.05, *t*-test) both in pig and cattle (Figure 2D), but significant differences between all the heterologous antibody titers and homologous antibody titer were observed in sheep. The mean antibody titer for heterologous virus SCGH/CHA/2016 with the greatest difference still reached 1.9 log10 (>1.65) in pig and sheep. These results were consistent with those obtained by NA-ELISA except that there was no significant difference between VN antibody titer for heterologous virus O/Tibet/99 and homologous virus in cattle.

### 3.3. Vaccine Matching Test Using Vaccinated Cattle Sera

Determination of indirect relationships (r1-value) between potential vaccine strains and field strains based on antibody responses against both are routinely used for vaccine matching purposes. The r1-value was calculated according to the cross-neutralization analysis results from serum samples collected from 13 cattle inoculated with the O/XJ/CHA/2017 vaccine (Group H) at 42 dpv. The r1-value between the O/XJ/CHA/2017 strain and other viruses, namely, O/Mya/98, SCGH/CHA/2016, and O/Tibet/99, was higher than 0.3 both using NA-ELISA (Figure 3A) and VNT (Figure 3B), suggesting that the O/XJ/CHA/2017 strain exhibited cross-reactivity with serotype O strains belonging to the representative topotypes/lineages circulated in China. According to r1-value determined by NA-ELISA, the antigen relationship between O/XJ/CHA/2017 strain and O/Mya/98 was closest, and SCGH/CHA/2016 was farthest (Figure 3A). Using VNT, similar trends in r1-values were observed, indicating that NA-ELISA could be used to determinate indirect relationships (r1-value) between potential vaccine strains and field strains.

### 3.4. Correlation between Antibody Titers against Each Topotype Virus in Vaccinated Animals Detected by NA-ELISA and VNT

A total of 139 sera collected from all vaccinated animals (One sheep in group L died during the test period) at 42 dpv were detected by NA-ELISA and by VNT for the FMDV O/Mya/98, FMDV SCGH/CHA/2016, FMDV O/Tibet/99, and FMDV O/XJ/CHA/2017 strains. A VNT was applied to measure specific neutralizing antibodies against FMDV to confirm whether the antibody detected by NA-ELISA was neutralizing antibody. The Pearson correlation coefficient between the results of NA-ELISA and those of VNT for each strain was calculated by comparing the data at the individual level. A statistically significant correlation between the NA-ELISA titers and the VNT titers was observed for all strains tested (Figure 4).

### 3.5. Relationship between NA-ELISA Titers of Vaccinated Pigs and Clinical Protection against FMDV Serotype O

The antibody titers against FMDV O/Mya/98 of 135 pig sera provided by vaccine manufacturer with known protective results were determined using NA-ELISA. The antibody titers and protection result for the individual animals were summarized in Figure 5. Seventy-six out of 77 pigs with titers >1.35 were protected against challenge (open squares), giving a protection rate of 99%. Nine out of 58 pigs with titers <1.35 were unprotected (filled triangles); thus, the protection rate was 84%. These results showed that the vaccinated pigs with NA-ELISA antibody positive could be protected, vaccinated pigs with NA-ELISA antibody negative were not necessarily unprotected, and the NA-ELISA titer was positively related to protection.

## 4. Discussion and Conclusions

FMDV is a highly variable RNA virus with seven immunological serotypes. The antigenic variability between and within serotypes limit the cross-reactivity, and therefore the in vivo cross-protection of vaccines, making it is crucial to select appropriate vaccine strain. As neutralizing antibody titers are closely related to protection in animals [16,17], they are widely used as the reference test system for vaccine strain selection. However, VNT also has the disadvantage of using live viruses and not being easy to scale up. In this study, we evaluated the reliability of NA-ELISA for vaccine strain screening. To be used as a vaccine strain, it must have good immunogenicity, thus the immunogenicity of FMDV O/Mya/98, SCGH/CHA/2016, O/Tibet/99, and O/XJ/CHA/2017 four field isolates was compared in pigs, cattle, and sheep. Both NA-ELISA and VNT results indicated that the O/XJ/CHA/2017 strain exhibited high immunogenicity in all vaccinated species (Figure 1). In this study, animals were vaccinated with vaccines containing a similar amount of antigen, offers an excellent opportunity to compare the immunogenicity. In the process of preparation of vaccine antigen, we found that the 146S antigen of the FMDV O/XJ/CHA/2017 strain was very stable compared with the other three strains, which may be one of the reasons for the good immunogenicity.

FMDV serotype O strains have a wide range of antigenic variation, it is preferable that vaccine strains have a broad antigen spectrum. Cross neutralization analysis results showed that O/XJ/CHA/2017 strain exhibited broad antigenic coverage based on experiments using serum samples from vaccinated pigs, sheep, and cattle both by NA-ELISA and VNT (Figure 2). Broad coverage was also observed in the vaccine matching test, O/XJ/CHA/2017 strain antigenically matched with representative circulating FMDV topotypes/lineages, including O/ME-SA/PanAsia, O/SEA/Mya-98, and O/Cathay in China, both using NA-ELISA and VNT (Figure 3). According to OIE guidelines, when using VNT, an r1-value equal to or greater than 0.3 indicates that the vaccine strain is likely to confer cross-protection against the field strain whereas, an r1-value less than 0.3 indicates a lack of such cross-protection. When using LPBE, an r1-value of 0.4 is considered indicative of a good vaccine match [18]. In this study, the r1-values between O/XJ/CHA/2017 strain and FMDV O/Mya/98, SCGH/CHA/2016, and O/Tibet/99 by VNT were 0.97, 0.86, and 0.93, respectively. When using NA-ELISA, the r1-values were 0.99, 0.71, and 0.87, respectively. It has been shown that r1-value estimations using low serum titer become less precise [19]; therefore, r1-value was determined with boost vaccination cattle sera in this study. As mentioned above, O/XJ/CHA/2017 strain has a close antigenic relationship with FMDV O/Mya/98, SCGH/CHA/2016, and O/Tibet/99 from the r1-value and can confer cross-protection against FMDV O/Mya/98, SCGH/CHA/2016, and O/Tibet/99. In fact, the antibody titer that relates with protection is not the same for different strains. A previous study that analyzed the serological results of cross-protection tests showed that r1-value was an inaccurate indicator of cross-protection regardless of the VNT titers of the samples [20]. Therefore, while the r1-value is important in evaluating whether the vaccine candidate strains have cross-protection against the field circulating strains, the antibody titer of the vaccine candidate strains against the field virus strains is more important.

One of the main goals of antibody titer analysis is to know whether a vaccine can protect against challenge. Due to ethical and economic reasons, 139 immunized animals (One sheep died during the test period) were not challenged in this study. To understand the relationship between the NA-ELISA titers and protection against FMDV challenge, a total of 135 sera with known protective results were used, only one pig with NA-ELISA antibody positive (NA-ELISA titer = 1.5) showed clinical signs of FMD, all pigs with high (>1.5) NA-ELISA titers were protected against FMDV challenge. Forty-nine out of 58 pigs with titers <1.35 were also protected, giving a protection rate of 84%, but protection rate was significantly lower than antibody positive pigs, suggesting that the NA-ELISA titer was positively related to protection. Vaccinated pigs with low or even no detectable levels of NA-ELISA antibodies were protected from FMDV challenge (Figure 5), cellular immune response and antibody affinity were probably involved in the protection. Similar results have been observed in previous experiments in our lab [21,22], some vaccinated pigs with no detectable levels of neutralizing antibodies could be protected from virulent FMDV challenge. This was also supported by other studies that animals with low levels of neutralizing antibodies could also be protected [23,24,25,26].

In conclusion, O/XJ/CHA/2017 exhibited high immunogenicity all in pigs, cattle, and sheep and was antigenically matched with representative circulating topotypes/lineages of FMDV serotype O (O/ME-SA/PanAsia, O/SEA/Mya-98, and O/Cathay) in China, suggesting that O/XJ/CHA/2017 could be used as a vaccine candidate. Data obtained with NA-ELISA were consistent with those obtained with VNT, indicating that NA-ELISA was comparable to VNT and could be used as the reference test system for vaccine strain selection.

## Figures and Tables

**Figure 1 vaccines-09-00387-f001:**
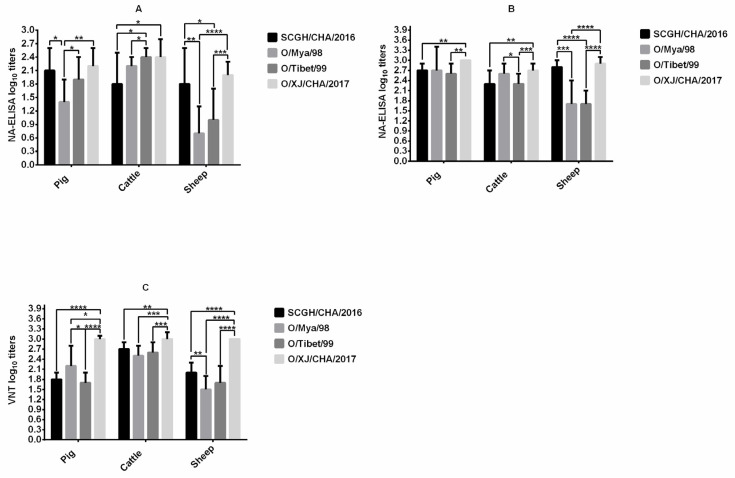
Comparison of immunogenicity in pigs, cattle, and sheep. Pigs (10 animals for each strain), cattle (12–13 animals for each strain) and sheep (12–13 animals for each strain) were immunized with Foot-and-mouth disease virus (FMDV) O/Mya/98, SCGH/CHA/2016, O/Tibet/99, and O/XJ/CHA/2017 vaccines at day 0 and 21, respectively. Serum samples collected at 21 days post vaccination (dpv) were used to measure homologous antibodies by NA-ELISA (**A**) and serum samples collected at 42 dpv with boosted vaccination were used to measure homologous antibodies both by NA-ELISA (**B**) and virus neutralizing test (VNT) (**C**). The statistical significance of differences between groups was determined. * *p* < 0.05, ** *p* < 0.01, *** *p* < 0.001, **** *p* < 0.0001.

**Figure 2 vaccines-09-00387-f002:**
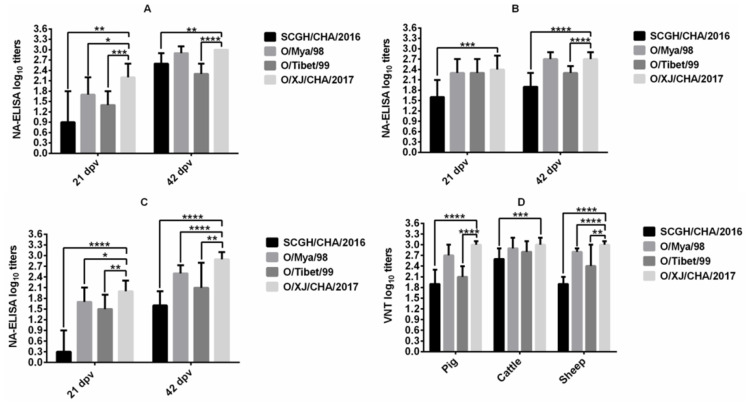
Cross-neutralization analysis. Serum samples collected at 21 dpv and 42dpv from pigs (**A**), cattle (**B**), and sheep (**C**) vaccinated with O/XJ/CHA/2017 vaccine were used to measure homologous antibodies with FMDV O/XJ/CHA/2017 and heterologous antibodies with FMDV O/Mya/98, SCGH/CHA/2016, and O/Tibet/99 by NA-ELISA, and sera collected at 42 dpv were also used to detect homologous and heterologous antibodies by VNT (**D**). Unpaired *t*-test was used to compare the differences between homologous antibodies and heterologous antibodies. * *p* < 0.05, ** *p* < 0.01, *** *p* < 0.001, **** *p* < 0.0001.

**Figure 3 vaccines-09-00387-f003:**
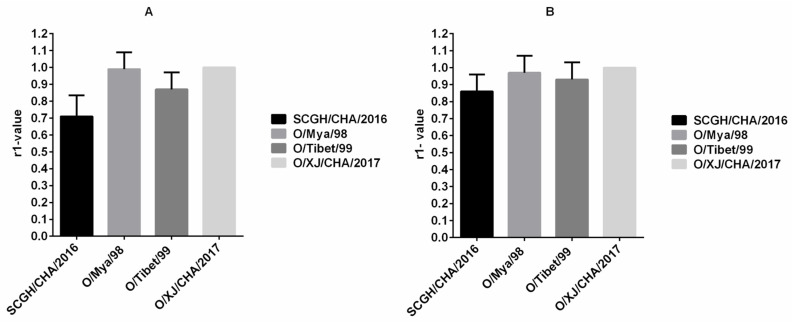
Vaccine matching test using vaccinated cattle sera. The r1-value between the O/XJ/CHA/2017 strain (homologous virus) and other viruses, namely, O/Mya/98, SCGH/CHA/2016, and O/Tibet/99 was calculated according to the cross-neutralization analysis results from serum samples collected from 13 cattle inoculated with the O/XJ/CHA/2017 vaccine at 42dpv both using NA-ELISA (**A**) and VNT (**B**).

**Figure 4 vaccines-09-00387-f004:**
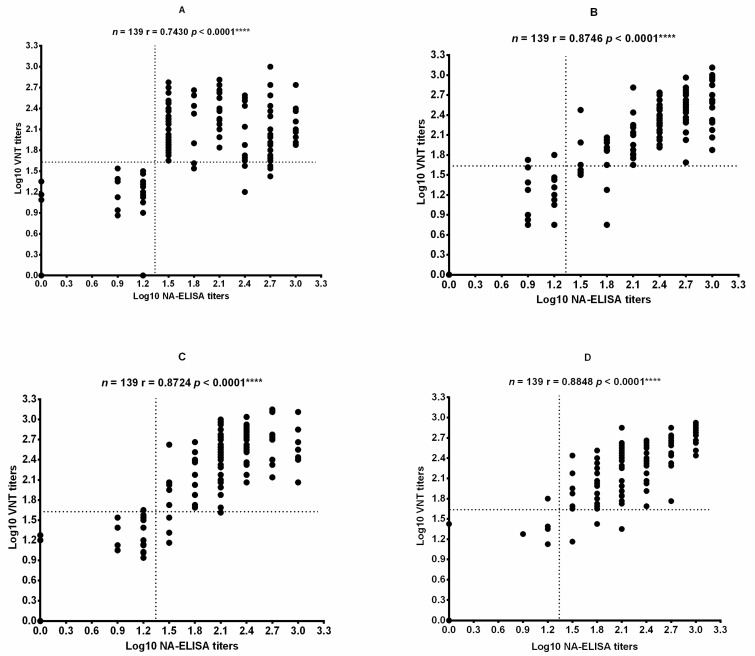
Correlation of the titers obtained by the NA-ELISA and VNT using different FMDV strains. (**A**) FMDV SCGH/CHA/2016. (**B**) FMDV O/Mya/98. (**C**) FMDV O/Tibet/99. (**D**) FMDV O/XJ/CHA/2017. Pearson’s correlation coefficient was computed using GraphPad Prism (version 6.0) software (San Diego, CA, USA). *n*, number of serum samples tested; r, correlation coefficient. The *P* value is two-tailed. The dotted lines represent the cutoff values. **** *p* < 0.0001.

**Figure 5 vaccines-09-00387-f005:**
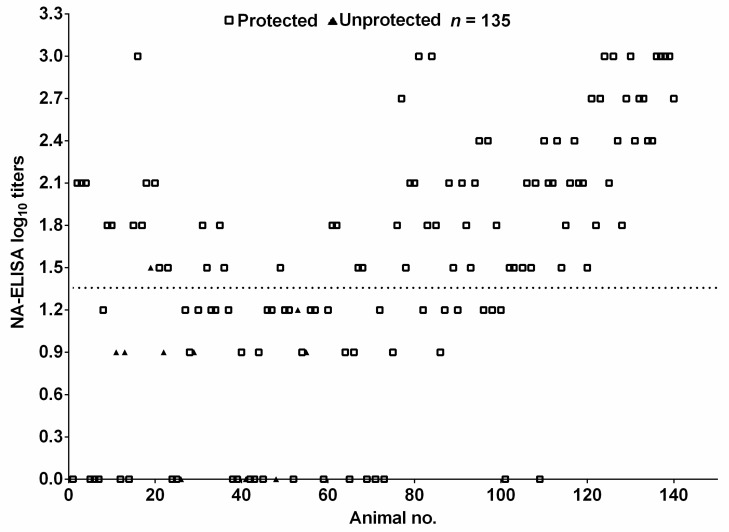
The relationship between NA-ELISA titers of vaccinated pigs and clinical protection against challenge with FMDV serotype O. Dispersion of individual values for protected pigs (open squares) and unprotected pigs (filled triangles). The dotted lines represent the cutoff value of NA-ELISA and *n*, number of pig sera tested.

**Table 1 vaccines-09-00387-t001:** Animal grouping and vaccine immunization.

Animal Species	Group	Vaccine Strain	No. of Animals	Vaccination Dose
Pigs	A	SCGH/CHA/2016	10	2 mL
	B	O/Mya/98	10	2 mL
	C	O/Tibet/99	10	2 mL
	D	O/XJ/CHA/2017	10	2 mL
Cattle	E	SCGH/CHA/2016	12	2 mL
	F	O/Mya/98	12	2 mL
	G	O/Tibet/99	13	2 mL
	H	O/XJ/CHA/2017	13	2 mL
Sheep	I	SCGH/CHA/2016	12	2 mL
	J	O/Mya/98	12	2 mL
	K	O/Tibet/99	13	2 mL
	L	O/XJ/CHA/2017	13	2 mL

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
