# Peer review of "Selection of Vaccine Candidate for Foot-and-Mouth Disease Virus Serotype O Using a Blocking Enzyme-Linked Immunosorbent Assay"

_vaccines, 2021, doi:10.3390/vaccines9040387_

Round 1

Reviewer 1 Report

Selection of Vaccine Candidate for Foot-and-mouth Disease Virus Serotype O using a Blocking Enzyme-Linked Immunosorbent Assay Based on Broadly Neutralizing Bovine Monoclonal Antibodies by Cao et al describes the selection of a vaccine candidate strain for use against serotype O FMDV currently circulating in China, and the authors evaluate the use of a recently developed NA-ELISA for vaccine candidate selection. The manuscript presents the study well, however several points should be addressed, as outlined below:

-Grammar and punctuation should be improved throughout the manuscript

-the title is long and does not address the aspect of evaluating NA-ELISA for vaccine selection

Section 2.2: describe the methods used to determine the correlation between NA-ELISA titers and clinical protection

Section 2.3: how were the viruses acquired? Describe how the viruses were grown (how many passages, etc.), and how virus stocks were prepared and stored.

Page 3 line 21: what is the unit for 28?

Section 2.8: why was only one group chosen to calculate r1 values? How did you choose Group H for this?

Page 5 line 2: remove “four” from this sentence

Page 8 line 2: “mechanically” doesn’t make sense; either explain what you mean or remove the word from this sentence and where it appears below

Section 3.5: it would be interesting to compare the NA-ELISA titer correlation with protection to a VNT titer correlation with protection. You make comparisons between NA-ELISA and VNT in all other sections.

Author Response

Reviewer 1

Selection of Vaccine Candidate for Foot-and-mouth Disease Virus Serotype O using a Blocking Enzyme-Linked Immunosorbent Assay Based on Broadly Neutralizing Bovine Monoclonal Antibodies by Cao et al describes the selection of a vaccine candidate strain for use against serotype O FMDV currently circulating in China, and the authors evaluate the use of a recently developed NA-ELISA for vaccine candidate selection. The manuscript presents the study well, however several points should be addressed, as outlined below:

-Grammar and punctuation should be improved throughout the manuscript

Response: Yes, we have improved grammar and punctuation throughout the manuscript and detailed changes are highlighted in red.

-the title is long and does not address the aspect of evaluating NA-ELISA for vaccine selection

Response: Yes, we have deleted “Based on Broadly Neutralizing Bovine Monoclonal Antibodies” and changed the title to “Selection of Vaccine Candidate for Foot-and-mouth Disease Virus Serotype O using a Blocking Enzyme-Linked Immunosorbent Assay”.

Section 2.2: describe the methods used to determine the correlation between NA-ELISA titers and clinical protection

Response: Yes, “the correlation” is not very exact, we have changed “the correlation” to “the relationship” and added the methods in the paper (page 3, lines 5-8).

Section 2.3: how were the viruses acquired? Describe how the viruses were grown (how many passages, etc.), and how virus stocks were prepared and stored.

Response: Yes, the viruses obtained from WRLFMD at the LVRI and were serially passaged two to three times in adherent BHK-21 cells. Stocks of virus were prepared by infecting BHK-21 cell monolayers and were stored as clarified tissue culture harvest material at -70 ◦C until required. We have added these contents in page 3, lines 18-21.

Page 3 line 21: what is the unit for 28?

Response: Yes, we have added “h” in page 3 line 25.

Section 2.8: why was only one group chosen to calculate r1 values? How did you choose Group H for this?

Response: Yes, r1-values were calculated using sera from cattle inoculated with the highest immunogenicity vaccine. Because the antibody response level of cattle is the highest, the r1-values are usually calculated using cattle. Therefore, Group H was chosen to calculate the r1-values in this experiment. We have modified the description in page 4 line 40.

Page 5 line 2: remove “four” from this sentence

Response: Yes, it was revised accordingly.

Page 8 line 2: “mechanically” doesn’t make sense; either explain what you mean or remove the word from this sentence and where it appears below

Response: Yes, it was revised accordingly.

Section 3.5: it would be interesting to compare the NA-ELISA titer correlation with protection to a VNT titer correlation with protection. You make comparisons between NA-ELISA and VNT in all other sections.

Response: Yes, it is really interesting to compare the NA-ELISA titer correlation with protection to a VNT titer correlation with protection. However, due to the limited amount of serum per serving, repeated neutralization tests have not been possible, but we will consider doing such tests in the future.

Reviewer 2 Report

The authors described that NA-ELISA, which was developed by themselves previously, could be used as a reference test system for vaccine strain selection through comparison with VNT and protection test using pig serum provided by vaccine manufacture. Then, they demonstrated that O/XJ/CHA/2017 was promising vaccine strain candidate. These findings are of considerable interest. A few minor revisions other than typographical errors are listed below.

  1. Title and Abstract. Although the title included bovine monoclonal antibodies, the authors didn't mention at all in the abstract. It seems to be unnecessary in the title.
  2. Page 3, line 21. 28 h.
  3. Table 1. Delete total number of animals.
  4. Page 4, lines 15-34. The authors should explain F128 and Bio-E46 for the benefit of the reader. 
  5. Page 6, lines 9-10. Delete this sentence.
  6. Page 10, lines 12-17. This sentence should be modified.

Author Response

Reviewer 2

The authors described that NA-ELISA, which was developed by themselves previously, could be used as a reference test system for vaccine strain selection through comparison with VNT and protection test using pig serum provided by vaccine manufacture. Then, they demonstrated that O/XJ/CHA/2017 was promising vaccine strain candidate. These findings are of considerable interest. A few minor revisions other than typographical errors are listed below.

  1. Title and Abstract. Although the title included bovine monoclonal antibodies, the authors didn't mention at all in the abstract. It seems to be unnecessary in the title.

Response 1: Yes, we have deleted “Based on Broadly Neutralizing Bovine Monoclonal Antibodies” and changed the title to “Selection of Vaccine Candidate for Foot-and-mouth Disease Virus Serotype O using a Blocking Enzyme-Linked Immunosorbent Assay”.

  1. Page 3, line 21. 28 h.

Response 2: Yes, it was revised accordingly.

  1. Table 1. Delete total number of animals.

Response 3: Yes, the total number has been deleted in the table 1.

  1. Page 4, lines 15-34. The authors should explain F128 and Bio-E46 for the benefit of the reader. 

Response 4: Yes, both F128 and E46 were bovine bnAbs against FMDV serotype O produced previously in our lab using a combination of fluorescence-activated cell sorter (FACS) and techniques for the isolation of single B cells. Bio-E46 was biotinylated bovine bnAbs E46. We have explained F128 and Bio-E46 in the paper (page 4, lines 21, 23 and 35-37).

  1. Page 6, lines 9-10. Delete this sentence.

Response 5: Yes, it was revised accordingly.

  1. Page 10, lines 12-17. This sentence should be modified.

Response 6: Yes, we have changed this sentence to “According to OIE guidelines, when using VNT, an r1-value equal to or greater than 0.3 indicates that the vaccine strain is likely to confer cross-protection against the field strain whereas, an r1-value less than 0.3 indicates a lack of such cross-protection. When using LPBE, an r1-value of 0.4 is considered indicative of a good vaccine match” in the paper (page 11, lines 4-8).

In addition, we found some minor errors in Figures 4 and 5, so we replaced them.